# EGR2, IGF1 and IL6 Expression Are Elevated in the Intervertebral Disc of Patients Suffering from Diffuse Idiopathic Skeletal Hyperostosis (DISH) Compared to Degenerative or Trauma Discs

**Benjamin Gantenbein** [1,2,*,†], **Rahel D. May** [1,†], **Paola Bermudez-Lekerika** [1,2], **Katharina A. C. Oswald** [2], **Lorin M. Benneker** [2] **and Christoph E. Albers** [2]

1. Tissue Engineering for Orthopaedics and Mechanobiology, Department for BioMedical Research (DBMR), University of Bern, CH-3008 Bern, Switzerland; Rayel.May@gmail.com (R.D.M.); paola.bermudez@dbmr.unibe.ch (P.B.-L.)
2. Department of Orthopaedic Surgery and Traumatology, Inselspital, University of Bern, CH-3010 Bern, Switzerland; katharina.oswald@insel.ch (K.A.C.O.); lorin.benneker@insel.ch (L.M.B.); christoph.albers@insel.ch (C.E.A.)
* Correspondence: benjamin.gantenbein@dbmr.unibe.ch or benjamin.gantenbein@insel.ch; Tel.: +41-31-632-88-15
† These authors contributed equally.

**Abstract:** Diffuse idiopathic skeletal hyperostosis (DISH) is characterised by ectopic ossification along the anterior spine and the outer intervertebral discs (IVD). However, the centre of the IVD, i.e., the nucleus pulposus, always remains unaffected, which could be due to the inhibition of the bone morphogenetic protein (BMP) pathway. In this study, we investigated the transcriptome for the BMP pathway of DISH-IVD cells versus disc cells of traumatic or degenerative discs. The disc cells originated from nucleus pulposus (NP), annulus fibrosus (AF) and from cartilaginous endplate (CEP) tissue. Here, ninety genes of the transforming growth factor β-BMP signalling pathway were screened by qPCR. Furthermore, the protein expression of genes of interest was further investigated by immune-staining and semi-quantitative microscopy. IVDs of three DISH patients were tested against three control patients (same disc level and similar age). Early Growth Response 2 (*EGR2*) and Interleukin 6 (*IL6*) were both significantly up-regulated in DISH-IVD cells compared to controls (12.8 ± 7.6-fold and 54.0 ± 46.5-fold, respectively, means ± SEM). Furthermore, Insulin-like Growth Factor 1 (IGF1) tended to be up-regulated in DISH-IVD donors, i.e., 174.13 ± 120.6-fold. IGF1 was already known as a serum marker for DISH and other rheumatoid diseases and is confirmed here to play a possible key role in DISH-IVD.

**Keywords:** low back pain; diffuse idiopathic skeletal hyperostosis; RNA extraction; qPCR; TGFβ-pathway; immune-histochemistry; Interleukin 6; Insulin-like growth factor 1; early growth response 2

## 1. Introduction

Diffuse idiopathic skeletal hyperostosis (DISH), also known as Forestier's disease, was first described by Forestier and Rotes-Querol in 1950 [1]. The authors characterised this disorder by spinal stiffness, osteophytosis, and aberrant new-bone formation along the spine's anterior lateral aspect. The underlying causes are yet unknown and thus are called "idiopathic" [2]. DISH is generally characterised by ossification of ligamentous insertions (fibro-osteosis) and ossification or calcifications of tendons, ligaments, fasciae, joint capsule and annulus fibrosus (AF) fibres [3–5]. Furthermore, DISH increases the amount of normal cancellous and cortical bone. Moreover, a tendency of post-operative heterotopic bone formation has been observed [6]. The incidence of DISH, however, remains unknown, but it seems that factors of metabolic, endocrine, genetic, and epigenetics seem to be important

in the development of this condition [4,7]. This disease mainly affects the elderly and is more frequent in men (65%) than in women, and the prevalence also increases with age (48–85) [8,9]. DISH patients often suffer from severe pain and need special treatment. Hence, the disease is often co-diagnosed with osteoarthritis [10,11].

Recently, Kuperus et al. (2016) [12] investigated the relationship between DISH and IVD degeneration. These authors found that partial or complete bone bridges were often present in DISH cadaveric IVDs that were only rarely found in healthy cadaveric controls. However, the disc height and degeneration level of DISH and control specimens were comparable. This specific feature also distinguishes the DISH condition from spondylosis [11].

To date, only very few molecular pathway studies have been conducted on the effect of DISH on IVDs. ElMiedany et al. (2000) proposed that the new formation of bone is mainly driven by abnormal cell growth/activity in the bony-ligamentous region [13]. Previous studies on the metabolomics of this disease found that patients suffering from symptomatic DISH disease have elevated levels of growth hormone (GH) and insulin-like growth factor 1 (IGF1) compared to patients of the same age and gender [9,14–19]. Thus, the molecular mechanism of DISH and the specific characteristics of underlying cell signalling remain obscure. Bone morphogenic proteins (BMPs), such as BMP2, BMP4, BMP6, BMP7, and BMP9, have shown their potential to induce the osteogenic differentiation of stromal cells and osteoblastic lineage cells in vitro and in vivo [20–23]. Furthermore, the recent literature has summarised and questioned the supra-physiological usage of BMP2 for the application of improved spinal fusion [24]. Thus, recently it has been hypothesised that BMP inhibitors could be the main reason for the lacking effect of BMP2 for improved spinal fusion [25–28]. In this context, it is of uttermost interest to analyse the transcriptome of DISH-IVDs in more detail, as all the surrounding tissue starts to ossify but not the IVD itself, which seems to be resistant to osteogenic signalling [29].

Hence, the current study aimed to compare the gene expression at the BMP pathway of cells of IVDs from patients suffering from DISH disease and compare it with cells originating from traumatic or degenerative IVDs. We assumed that trauma discs, especially from younger patients, contain presumably "healthier" cells that could be defined as a "control". Here, we investigated whether DISH disease is linked to alterations in BMP/TGF$\beta$ signalling pathways to gain further insights into how the DISH-IVD was affected by this idiopathic disease at the molecular level. We hypothesised that IVDs of patients diagnosed with DISH demonstrate specific gene expression changes affecting the TGF$\beta$/BMP pathway compared to degenerative IVDs or trauma patients.

## 2. Materials and Methods

### 2.1. IVD Donor Materials and Cell Isolation

Human IVDs were obtained from patients suffering from DISH disease. As controls, traumatic or degenerative IVDs of patients undergoing spinal surgery were used (Table 1). Patients gave their written consent, and procedures were approved by the ethics committee of the Canton of Bern. Tissues were processed within 24 h after surgery.

Human IVD tissue was separated into nucleus pulposus (NP), annulus fibrosus (AF) and cartilaginous endplate (CEP) tissue by an experienced surgeon or processed as mixed IVD. Cell isolation from the native extracellular matrix was performed as described earlier [28]. Briefly, tissue was enzymatically digested by 1 h in pronase (Roche, Basel, Switzerland) followed by overnight incubation in collagenase type 2 (Worthington, London, United Kingdom). Subsequent expansion of NP cells (NPC), AF cells (AFC), and CEP cells (CEPC) followed in proliferation medium (low-glucose (1g/L) Dulbecco's Modified Eagle Medium (LG-DMEM) supplemented with 10% FBS and 1% P/S).

IVD cells were obtained from six DISH donors aged from 65 to 84 years (72.3 $\pm$ 3.1 years) and three "control" donors, which were lacking symptoms of DISH, suffering from degenerated or traumatic discs aged from 17 to 66 years (45 $\pm$ 14.5 years).

**Table 1.** Donor list of intervertebral disc cells. Cell type:nucleus pulposus (N), annulus fibrosus (A), cartilaginous endplate (CEP), C = cervical, L = lumbar, intervertebral disc mixed (IVD) Sex: Female (F), Male (M). Disc condition: Degenerated (D), Trauma (T), Pfirrmann Grade (PG).

| Donor Groups Used in this Study | | | | | | | | | | |
|---|---|---|---|---|---|---|---|---|---|---|
| **DISH** | | | | | **Control** | | | | | |
| Patient # | Cell Type | Age | Sex | Spinal Level | Patient # | Cell Type | Age | Sex | Spinal Level | Disc Condition |
| 1 | N/A/CEP | 72 | M | L4/L5 | 1 | N/A/CE | 66 | F | L4/L5 | D; PG3 |
| 2 | N/A/CEP | 65 | F | L11/T12 | 2 | N/A/CE | 52 | M | L11/T12 | T; PG3 |
| 3 | IVD | 66 | M | L5/L6 | 3 | IVD | 17 | M | C5/C6 | T |
| 4 | N/A | 68 | M | L11/T12 | 2 | N/A/CE | 52 | M | L11/T12 | T; PG3 |
| 5 | A/CEP | 79 | M | C5/C6 | 3 | IVD | 17 | M | C5/C6 | T |
| 6 | A | 84 | M | L10/T11 | 2 | N/A/CE | 52 | M | L11/T12 | T; PG3 |

*2.2. Analysis of Specific Gene Expression in Human Primary IVD Cells with Quantitative Polymerase Chain Reaction (qPCR)*

Cells were cultured at passage 1–2 (P1–2) in monolayer, and then trypsinised before 2 Mio of cells were lysed directly with TRI. The extraction of total RNA was performed as described in the methods before [28,30]. Prior qPCR RNA integrity and purity were checked on selected samples by the Experion™ Automated Electrophoresis System (Bio-Rad, Reinach, Switzerland). Any possible remaining DNA was digested by using DNase I (AMP-D1, Sigma-Aldrich, Buchs, Switzerland) for 15 min. Reverse transcription was performed by using all-in-one cDNA Synthesis SuperMix (Bimake distributed by LuBio-Science GmbH, Lucerne, Switzerland). Real-time PCR was performed using SYBR Green PCR master mix on a CFX96touch qPCR system (all from Bio-Rad). Ninety genes (Table 2) were tested with PrimePCR™ Pathway plate 96 well (PrimePCR™ TGFβ BMP Signalling Pathway Plus H96, 90 genes, Bio-Rad, US). The qPCR was run using a two-step protocol with an annealing temperature of 60 °C for 30 s and 95 °C for 5 s for 40 cycles (according to the recommendations of the manufacturer). Amplicon specificity was ensured by performing a melting curve analysis after cycling. CFX manager software version 3.1 (Bio-Rad) was used for the quantification analysis of normalised relative gene expression. The number of PCR cycles needed for each sample to reach the threshold level was recorded as the C$q$ value [31]. Values were normalised relative to two reference genes (18S and ACTB). DISH and control cells were compared in scatter plots and RNA transcripts that differed more than four times were indicated in red (up-regulated in DISH patients) or in green (down-regulated in DISH patients).

**Table 2.** List of genes tested in PrimePCR™ TGFβ BMP Signalling Pathway Plus H96, 90 genes, Bio-Rad, US.

| Name | Description | Name | Description | Name | Description |
|---|---|---|---|---|---|
| *ACTB* | Actin beta | *EMP1* | Epithelial Membrane Protein 1 | *NOG* | Noggin |
| *ACTC1* | Actin Alpha Cardiac Muscle 1 | *ENG* | Endoglin | *NOV* | Nephroblastoma overexpressed |
| *ACVR1* | Activin A Receptor Type 1 | *FAS* | Fas Cell Surface Death Receptor | *PDGFB* | Platelet-Derived Growth Factor Subunit B |
| *ACVR2A* | Activin A Receptor Type 2A | *FGF2* | Fibroblast Growth Factor 2 | *PLAU* | Plasminogen Activator, Urokinase |
| *ACVRL1* | Activin A Receptor-Like Type 1 | *FSTL3* | Follistatin Like 3 | *PMEPA1* | Prostate Transmembrane Protein, Androgen Induced 1 |

**Table 2.** *Cont.*

| Name | Description | Name | Description | Name | Description |
|---|---|---|---|---|---|
| AMH | Anti-Mullerian Hormone | GADD45B | Growth Arrest and DNA Damage Inducible Beta | RPLP0 | Ribosomal Protein Lateral Stalk Subunit P0 |
| AMHR2 | Anti-Mullerian Hormone Receptor Type 2 | GAPDH | Glyceraldehyde-3-Phosphate Dehydrogenase | SERPINE1 | Serpine Family E Member 1 |
| ATF4 | Activating Transcription Factor 4 | GDF2 | Growth Differentiation Factor 2 | SMAD1 | SMAD Family Member 1 |
| B2M | Beta-2-Microglobulin | GDF3 | Growth Differentiation Factor 3 | SMAD2 | SMAD Family Member 2 |
| BAMBI | BMP and Activin Membrane Bound Inhibitor | GDF5 | Growth Differentiation Factor 5 | SMAD3 | SMAD Family Member 3 |
| BGLAP | Bone Gamma-Carboxyglutamate Protein | GDF6 | Growth Differentiation Factor 6 | SMAD4 | SMAD Family Member 4 |
| BMP1 | Bone Morphogenetic Protein 1 | GDF7 | Growth Differentiation Factor 7 | SMAD6 | SMAD Family Member 6 |
| BMP2 | Bone Morphogenetic Protein 2 | GSC | Goosecoid Homeobox | SMAD7 | SMAD Family Member 7 |
| BMP3 | Bone Morphogenetic Protein 3 | GUSB | Glucuronidase Beta | SMURF1 | SMAD Specific E3 Ubiquitin Protein Ligase 1 |
| BMP4 | Bone Morphogenetic Protein 4 | HERPUD1 | Homocystein Inducible ER Protein with Ubiquitin Like Domain 1 | SOX4 | SRY-Box 4 |
| BMP5 | Bone Morphogenetic Protein 5 | HPRT1 | Hypoxanthine Phosphoribosyl-transferase 1 | STAT1 | Signal Transducer and Activator of Transcription 1 |
| BMP6 | Bone Morphogenetic Protein 6 | ID1 | Inhibitor of DNA Binding 1, HLH Protein | STK38L | Serine/Threonine Kinase 23 Like |
| BMP7 | Bone Morphogenetic Protein 7 | ID2 | Inhibitor of DNA Binding 2 | TBP | TATA-Box Binding Protein |
| BMPER | BMP Binding Endothelial Regulator | IFRD1 | Interferon Related Developmental Regulator 1 | TGFB1 | Transforming Growth Factor Beta 1 |
| BMPR1A | Bone Morphogenetic Protein Receptor, Type IA | IGF1 | Insulin-Like Growth Factor 1 | TGFB1l1 | Transforming Growth Gactor Beta-1-induced Transcript 1 |
| BMPR1B | Bone Morphogenetic Protein Receptor, Type IB | IGFBP3 | Insulin-Like Growth Factor Binding Protein 3 | TGFB2 | Transforming Growth Factor Beta 2 |
| BMPR2 | Bone Morphogenetic Protein Receptor, Type II | IL6 | Interleukin 6 | TGFB3 | Transforming Growth Factor Beta 3 |

**Table 2.** *Cont.*

| Name | Description | Name | Description | Name | Description |
|---|---|---|---|---|---|
| CDKN1A | Cyclin-Dependent Kinase Inhibitor 1A | INHA | Inhibin Subunit Alpha | TGFBI | Transforming Growth Factor Beta Induced |
| CDKN1B | Cyclin-Dependent Kinase Inhibitor 1B | JUNB | JunB Proto-Oncogene, AP-1 Transcription Factor Subunit | TGFBR1 | Transforming Growth Factor Beta Receptor 1 |
| CDKN2B | Cyclin-Dependent Kinase Inhibitor 2B | KANK4 | KN Motif and Ankyrin Repeat Domains 4 | TGFBR2 | Transforming Growth Factor Beta Receptor 2 |
| CHRD | Chordin | KLHL24 | Kelch Like Family Member 24 | TGFBR3 | Transforming Growth Factor Beta Receptor 3 |
| COL1A1 | Collagen Type I Alpha 1 Chain | LEFTY1 | Left-Right Determination Factor 1 | TGIF1 | TGFB Induced Factor Homeobox 1 |
| COL1A2 | Collagen Type I Alpha 2 Chain | LTBP1 | Latent Transforming Growth Factor Beta Binding Protein 1 | THBS1 | Thrombospondin 1 |
| DCN | Decorin | MYC | MYC Proto-Oncogene, BHLH Transcription Factor | TNFSF10 | TNF Superfamily Member 10 |
| EGR2 | Early Growth Response 2 | NODAL | Nodal Growth Differentiation Factor | UBASH3B | Ubiquitin Associated and SH3 Domain Containing B |

### 2.3. Statistical Analysis and Visualisation of Heat Maps and Clustering

Relative gene expression was quantified between DISH and "control" patients. Statistical significance was tested using a non-parametric Mann–Whitney U test due to relatively low sample sizes using Prism 7.0d for Mac OS X (GraphPad, La Jolla, CA, USA). Non-parametric distribution was assumed. Consequently, statistical significance was determined using the Mann–Whitney U test. A $p$-value < 0.05 was considered to be significant. Relative gene expression data were normalised using CFX manager software 3.1. Heat mapping and hierarchical clustering was performed in CFX manager based on the degree of similarity of expression for different samples, and a heat map was generated alongside as implemented in CFX manager software. Pairwise scatter plot analysis was performed between DISH tissues versus "control" tissues and relative gene expression differences > 4.0 were highlighted (Supplementary Figure S1).

### 2.4. Immunocytochemistry

Cells at P1/P2 of frozen DISH donor 1 and control donor 1 (Table 1) were thawed and seeded at a density of 8000 cells/mL in glass chamber slides with removable 12-well silicone chambers for immunofluorescence staining (growth area per well 1.9 cm$^2$) (Vitaris, Baar, Switzerland). Cells were then cultured in LG-DMEM supplemented with 10% FBS and 1% P/S until they reached 70% confluency. Cells were then fixed in 4% buffered paraformaldehyde (Sigma-Aldrich) for 15 min and stored in 70% ethanol at 4 °C prior to staining. Fixed cells were permeabilized with 0.2% Triton (Sigma-Aldrich) for 15 min. Again, the cells were washed three times with Tris-buffered saline (TBS) for 5 min. Subsequently, cells were blocked with 10% FCS in 0.025% Tris-buffered saline

containing Tween 20 (TBS-T) for 30 min at room temperature. After blocking, cells were incubated overnight at 4 °C using a primary antibody (AB) in blocking solution and under continuous agitation (see Table 3). Secondary ABs were applied in 1:200 dilution in 0.025% TBS-T for 3 h at room temperature (Alexa Fluor® 555 Rabbit Anti-Goat IgG (H+L), Alexa Fluor® 555 Goat anti-mouse SFX-kit, Alexa Fluor® 488 Goat anti-Rabbit SFX Kit, Molecular Probes, Life Technologies, Basel, Switzerland). Cells were viewed under a microscope (Leica DMI4000 (Biosystems inc., Muttenz, Switzerland) and LAS AF Software).

**Table 3.** Primary antibodies used for immunohistochemistry.

| Primary AB | Source | Concentration ICC |
|---|---|---|
| Mouse GDF5 | Polyclonal Goat IgG (Biotechne, Zug, Switzerland) | 5 µg/mL |
| Human IL6 | Monoclonal mouse IgG (Biotechne) | 16 µg/mL |
| Human IGF1 | Polyclonal Goat IgG (Biotechne) | 5 µg/mL |
| Human BMP2 | Polyclonal Rabbit (Novusbio, Zug, Switzerland) | 5.75 µg/mL |
| Human EGR2 | Polyclonal Rabbit (Novusbio) | 5 µg/mL |
| Human GDF6 | Rabbit polyclonal–C terminal (Abcam, Cambridge, UK) | 1:100 |

## 3. Results

### 3.1. Prime PCR of DISH and Traumatic/Degenerative Discs

PrimePCR™ screening for 90 genes of the BMP pathway was initially run. Most strikingly, these data revealed a significant mean up-regulation of *EGR2* (12.8 ± 7.6-fold, $p = 0.0068$, overall mean ± SEM of all comparisons) and of interleukin 6 (*IL6*) (54.0 ± 46.5-fold, $p = 0.0005$) in DISH-IVDs relative to the "control" discs (Figure 1, Supplementary Figure S1). Furthermore, *IGF1* tended to be up-regulated in DISH-IVD donors (174.1 ± 120.6-fold, $p = 0.1704$), although this was non-significant. Growth and Differentiation Factor 5 (*GDF5*) and *GDF6* both tended to be down-regulated in the DISH-IVDs (i.e., −11.5 ± 10.0, $p = 0.26$ and −3.7 ± 3.1 $p = 0.30$, respectively, Figure 1, Supplementary Figure S1). However, these differences were not significant.

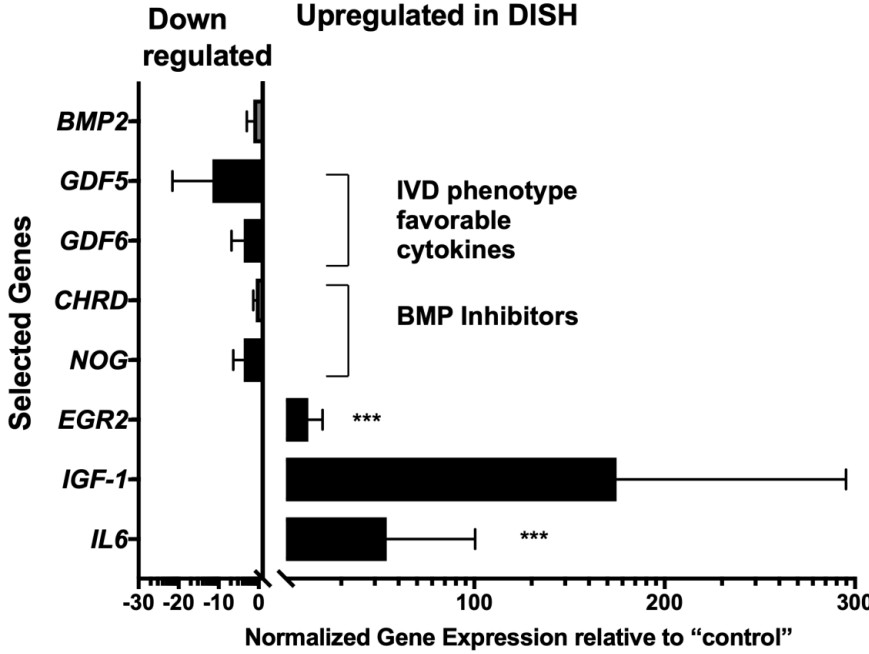

**Figure 1.** Normalised relative gene expression in DISH-IVDs relative to IVDs of "control" patients. The relative gene expression of IL6 and EGR2 was significantly up-regulated: *** $p < 0.01$. The scatterplots of the pairwise comparisons of the 90 PrimePCR™ gene array is given in Supplementary Figure S1. Means ± SEM.

Clustering the gene expression profiles of the 19 analysed samples of the three control patients and the six DISH patients revealed that not all samples clustered together as expected (Figure 2). However, it seems also evident that the hierarchical phenogram generated two main clusters of phenotypes with a similar TGFβ pathway pattern (Figure 2). Clade 1 consisted of mainly DISH samples, and clade 2 consisted mainly of control samples. However, samples of DISH patients 3 and 5 (i.e., D P3 IVDC and D P5 AFC and EPC) showed a phenotype more similar to the cells of the included trauma patients. On the other hand, the dendrogram confirmed that cells originating from different tissue types of the IVD from the same patient were grouped closely together, which was expected.

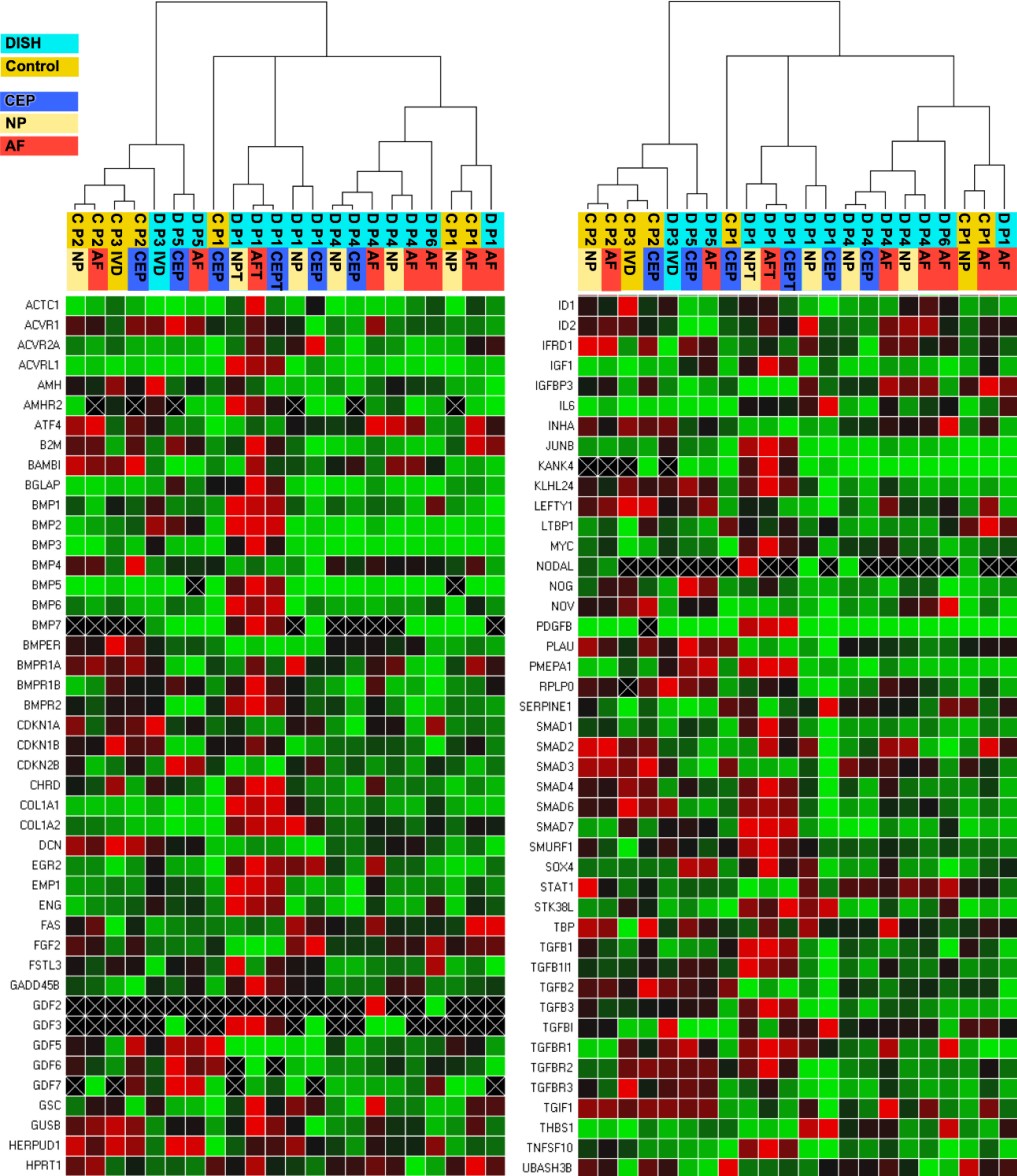

**Figure 2.** Clustergram displaying hierarchy based on the degree of similarity of expression for different samples and normalised gene expression heat map of relative gene expression at 88 genes of PrimePCR™ of the TGFβ-BMP Signalling Pathway Plus H96. Red indicates up-regulations, green indicates down-regulations and black indicates no regulations. The lighter the shade of colour, the greater the relative normalised expression difference. If no normalised expression value could be calculated, this is shown in black with a white X. Relative gene expression was estimated using ACTB and GAPDH as the two reference genes. Samples encoded in orange are samples categorised as trauma patients = control (C), and samples in light blue are from patients that were diagnosed with DISH (D). Tissue types were colour-coded in dark blue for CEP, in yellow for NP, and in red for AF.

### 3.2. Immunocytochemistry of IGF1, IL6, EGR2, BMP2, GDF5, and GDF6

Gene expression differences, which showed a more than four times up- or down-regulation in PrimePCR™ between DISH and "control" patients (Figure 1), were further evaluated using immune-staining of isolated cells grown in monolayer. BMP2 showed a stronger signal in the AFC in DISH than in "control" patients, as found in the relative gene expression (Figure 3). Furthermore, the EGR staining was more intense in the AFC and NPC in DISH compared to control patients' staining, as was also found in PrimePCR™. In particular, the AFC showed a strong immunocytochemistry staining for EGR2 and BMP2 compared to the AFC of the control patient. Additionally, IGF1 staining of DISH was more intense compared to control patients, as the NPC and the AFC showed almost no IGF1 protein expression by immunocytochemistry staining. GDF5 and GDF6 were overall more strongly stained in control cells compared to DISH cells.

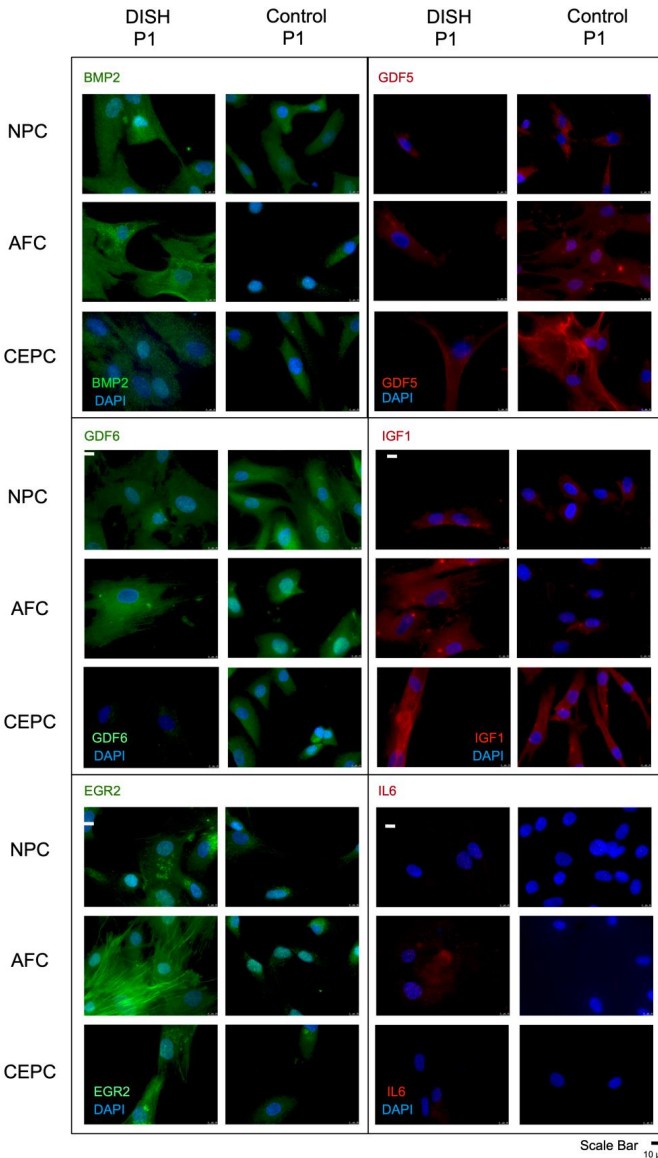

**Figure 3.** Fluorescence microscopy of immune-histochemistry of nucleus pulposus (NPC), annulus fibrosus (AFC) and cartilaginous endplate cells (CEPC) of DISH versus "control" patient 1 using human GDF5, human IL6, human IGF1, human BMP2, and human EGR2, and human GDF6 antibody and the secondary antibody labeled with the fluorescent dye Alexa 555 (red), and Alexa 488 (green), respectively.

## 4. Discussion

In this study, we report, for the first time, on the phenotypic changes in IVD cells from DISH patients compared to IVD cells from supposedly "healthier" cells from younger and mid-aged trauma patients (Table 1). We observed the consistent up-regulation of IGF-1, and EGR2, which seems to be a hallmark of the DISH-IVD cells [4,29], and cells from degenerative discs showed a different BMP gene-related transcript and protein expression (Figures 1–3). One of the main limitations of this study was the relatively small sample size. Thus, sample collection from a single hospital was a challenge, and future studies should recruit IVD material from bigger cohorts and multiple populations and etiologies. Additionally, the "controls" used in the study were not truly healthy IVDs, as they were either degenerative or traumatic discs. In this respect, a comparison of fresh cadaveric IVD material of undegenerated nature would be essential for any future investigations. Acquiring more DISH and matching healthy IVDs are required to support our first findings regarding the phenotypic status of the IVD during DISH disease. Moreover, it should be investigated whether DISH affects the overall phenotype of IVD cells and their tissues. To answer this particular question, complete transcriptome analyses using bulk RNA next-generation sequencing would definitely shed new light on the better understanding of DISH pathology and into yet-unidentified modified pathways of this disease.

Since the discovery of DISH, various studies, especially concerning the metabolic conditions of these patients, were conducted. Nevertheless, this disease is still poorly understood. In the current study, we investigated the changes at the transcript and the protein expression level of IVDs from patients diagnosed with DISH. We observed significant up-regulation of EGR2 and IL6 and a trend of higher IGF1 expression at the transcript level. These changes could be confirmed qualitatively at the protein level using immunohistochemistry. For GDF5 and GDF6, we noticed, for both genes, a down-regulation in gene expression in the IVD of those patients, though this was not statistically significant. However, these differences were not apparent at the protein level (Figure 3). This gene expression profile points to an involvement in inflammation in DISH, recently identified in a review by Mader et al. (2021) [29].

Previous research by Levi et al. (1996) [32] has shown that EGR2-/- mice develop skeletal abnormalities, i.e., reducing length and thickness in newly formed bone and calcified trabeculae. These phenomena were explained by the activation of EGR2 in a subpopulation of hypertrophic chondrocytes of the growth plate and differentiating osteoblasts (OBs) [32]. Furthermore, Leclerc et al. (2008) [33] showed up-regulation of EGR2 expression during the late stage of OB differentiation in-vitro. Furthermore, Zaman et al. (2012) [34] found that EGR2 expression was up-regulated in the osteosarcoma-derived UMR106 cell line through IGF1 stimulation. The cause of the observed up-regulation of EGR2 and its role in DISH-IVD cells would need to be further explored.

DISH has been associated with the change in inflammatory mediators [35]. Interleukin 1 (IL1) and IL6 are increased in DISH patients through the activation of nuclear factor kappa-B (NF-κB) ligand, and these cytokines stimulate the proliferation of OBs and bone deposition [29,35,36]. Two other studies [37,38] have also suggested the stimulating effect of IL6 on bone formation and turnover. Kang et al. (1996) [39] further demonstrated an increased concentration of IL6 in herniated lumbar discs, which may indicate an association with degradation of the IVD. We could most interestingly confirm the involvement of IL6 in our data, as it was up-regulated in all analysed DISH-IVDs (Figure 1, and Supplementary Figure S1).

Several studies have shown the possible overlap between DISH, obesity and Diabetes mellitus, with a pathogenic effect of insulin, IGF1 and GH [40–43]. Yakar et al. (2002) [44] demonstrated an association of circulating blood serum IGF1 and bone mass in mice with reduced IGF1 serum levels. These mice showed abnormalities in bone formation including decreased bone mineral density and cortical thickness [44]. However, IGF1 is not only present in the serum but also readily available in the skeletal environment. The skeletal homeostasis is driven by the release of IGF1 during bone modelling and remodelling, as

an essential mitogen and chemotactic player in the skeleton [45]. It also could have been shown that IGF1 stimulates IVD cell proliferation and matrix synthesis in-vitro [46].

GDF5, GDF6, and BMP2 are members of the TGFβ superfamily, with GDF5 and GDF6 being involved in the growth and development of cartilage and bone tissues [47]. Moreover, in the IVD, GDF5 seems to play a role in altering homeostasis, as several studies showed its beneficial effect of inducing cell proliferation, proteoglycan stimulation and COL2 expression [48,49]. Furthermore, GDF6 plays a role in spinal column development, especially in IVD development and homeostasis [50,51]. Both GDF5 and GDF6 have recently been shown to promote mesenchymal stromal cells towards a discogenic phenotype [52–56]. Interestingly, both GDF5 and GDF6 were down-regulated in several cell types of DISH-IVDs compared to controls. This might indicate a DISH-related change of the discogenic phenotype in these patients.

With increasing rates of obesity and type 2 Diabetes, the prevalence of DISH is also expected to rise in the future, possibly also affecting younger patients below 50 years of age [29,57]. This development is leading to an urgency to further understand and investigate this disease and its cell signalling.

## 5. Conclusions

- DISH-IVD, in contrast to IVD obtained from trauma, showed an up-regulation of *EGR2* and *IL-6*, and *IGF-1* tended to be up-regulated.
- Dysregulation of IGF-1 has been proposed for DISH-patients to be a serum-specific marker, and our data from IVD tissue pointed in this direction [4].
- *GDF5* and *GDF6* tended to be down-regulated in DISH-IVD compared to trauma-IVD.
- The observed differences in gene expression and protein differences could also be a result of factors other than DISH such as age and gender bias [58].

**Supplementary Materials:** The following are available online at https://www.mdpi.com/article/10.3390/app11094072/s1: Figure S1: TGFβ pathway transcriptome analysis of three donors with age- and disc level matched controls.

**Author Contributions:** B.G. and R.D.M. were responsible for the conception and design of the study. R.D.M. performed the experiments, data analysis and prepared the figures. R.D.M. drafted the manuscript; B.G. did extensive editing and prepared all of the figures; P.B.-L. and K.A.C.O. reviewed and edited the manuscript; C.E.A. and L.M.B. provided donor materials and clinical input and contributed to editing. All authors participated in the interpretation of the findings and approved the final version of the manuscript. B.G. and C.E.A. provided funding. All authors have read and agreed to the published version of the manuscript.

**Funding:** This work was supported from a start-up grant from the Center for Applied Biotechnology and Molecular Medicine (CABMM) to C.E.A. and B.G, and by funds from the Marie Skłodowska Curie International Training Network (ITN) "disc4all" (https://cordis.europa.eu/project/id/955735, accessed on 28 April 2021). Further funds were received from the clinical trials unit of Bern University Hospital and by a Eurospine Task Force Research grant #2019_22.

**Institutional Review Board Statement:** Not applicable.

**Informed Consent Statement:** All primary human cells presented in this article were obtained with written consent from the donors. The study was conducted in accordance with the Declaration of Helsinki, and the Ethics Committee approved the protocol of the Canton of Bern (Ref. 2019-00097).

**Data Availability Statement:** All data presented in this manuscript can be obtained upon request from the corresponding author.

**Acknowledgments:** We thank Andrea Oberli, Selina Steiner, and Eva Roth for laboratory assistance. The microscopes were provided by the microscope core facility of the University of Bern (www.mic.unibe.ch, accessed 28 April 2021).

**Conflicts of Interest:** The authors declare no conflict of interest.

## Abbreviations

| | |
|---|---|
| AB | Antibody |
| AF | Annulus fibrosus |
| AS | Ankylosing spondylitis |
| BMP | Bone morphogenetic protein |
| BMP2 | Bone morphogenetic protein 2 |
| BSA | Bovine serum albumin |
| C | Cervical |
| CEP | Cartilaginous endplate |
| D | Degenerated |
| DISH | Diffuse idiopathic skeletal hyperostosis |
| EGR2 | Early growth response 2 |
| GDF5 | Growth and differentiation factor 5 |
| GDF6 | Growth and differentiation factor 6 |
| GH | Growth hormone |
| IGF1 | Insulin-like growth factor 1 |
| IL1 | Interleukin 1 |
| IL6 | Interleukin 6 |
| IVD | Intervertebral disc |
| L | Lumbar |
| LG-DMEM | Low-glucose Dulbecco's Modified Eagle Medium |
| NP | Nucleus pulposus |
| OB | Osteoblast |
| TBS-T | Tris-buffered saline containing Tween 20 |
| TGF-β | Transforming growth factor β |
| P | Passage |
| PG | Pfirrmann grade |
| qPCR | Quantitative real-time Polymerase Chain Reaction |
| T | Trauma |

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
