# Peer review of "EGR2, IGF1 and IL6 Expression Are Elevated in the Intervertebral Disc of Patients Suffering from Diffuse Idiopathic Skeletal Hyperostosis (DISH) Compared to Degenerative or Trauma Discs"

_applsci, doi:10.3390/app11094072_

Round 1

Reviewer 1 Report

Thank you for the opportunity to review this publication. I believe that the presented research idea was broadly developed before,
and now there is an attempt to refresh it. The bibliography should be updated, as only a few of the 54 publications
used are under 10 years old. The clinical purpose of this study has not been well described. I do not consider it correct to use several self-citations in different
author configurations. I believe that the number of respondents should be higher,
such a small number of patients affects the quality of the results.

Author Response

1.1 Thank you for the opportunity to review this publication. I believe that the presented research idea was broadly developed before, and now there is an attempt to refresh it. The bibliography should be updated, as only a few of the 54 publications
used are under 10 years old. The clinical purpose of this study has not been well described. I do not consider it correct to use several self-citations in different author configurations. I believe that the number of respondents should be higher, 
such a small number of patients affects the quality of the results. 

Answer: We thank the reviewer for the comment. Of course, we are aware of the limited number of DISH cases, which we have analyzed so far. It is also questionable whether the “control” can be called a control group at all, as we were lacking clinical material from truly healthy non-degenerated or non-trauma patients. However, as the law in Switzerland on transplantation is different from other European countries, it is very difficult if not impossible to get access to fresh non-degenerated IVD tissue form younger donors. This is why we decided to focus on trauma or only minor degenerated Ivd material as the control group. We have stated this clearly in the manuscript. Nevertheless, analysing in more detail the currently available material we found increased expression/de-regulation of certain genes in the BMP/TGFβ-qPCR array (90 genes), and also semi-quantitatively by analyzing cells for immune-histochemistry at selected markers. Thus, we cannot address the sample size within this revision. However, we addressed the concerns not having considered newer citations. Thus, we have improved the introduction, we also added important citations, and updated the bibliography accordingly. We deleted some of the self-citations, and kept this to a minimum as suggested and needed in the citation of previously established protocols.

Reviewer 2 Report

This paper describes the results of the study of intervertebral disks of patients with DISH vs. controls.

Overall, the paper is clear and the conclusions are well supported.

Comments for improvement:
The introduction is quite long. While it gives useful and important background on DISH and its history of pathophysiology, I believe it could be curtailed significantly.

Why it is important to study the IVD, when DISH affects other structures around the spine, has to be better substantiated. In particular, please expand on why studying the IVD may help you better understand the molecular mechanism of DISH and the specific characteristics underlying cell signaling.

Why are the Materials and Methods section (section 4) reported after Results and Discussion? Please move them to section 2 and move the results and discussion down correspondingly.

The discussion section is also quite lengthy, and repetitious in parts. For example, lines 8-9 are somewhat repeated in lines 12-13 (page 7).

Author Response

Reviewer 2

This paper describes the results of the study of intervertebral disks of patients with DISH vs. controls.

2.1 Overall, the paper is clear and the conclusions are well supported.

Answer: We thank the reviewer for the comment.

2.2 The introduction is quite long. While it gives useful and important background on DISH and its history of pathophysiology, I believe it could be curtailed significantly.

Answer: We revised the introduction and are focusing only on DISH and we cut-down the relatively lengthy introduction as requested by reviewer 1. However, we made sure also to consider the requested detail on the TGFb pathway as requested from reviewer 3 (see our answer to 3.3.) , and at the same time cut-down the history of DISH to keep the introduction more compact.

2.3 Why it is important to study the IVD, when DISH affects other structures around the spine, has to be better substantiated. In particular, please expand on why studying the IVD may help you better understand the molecular mechanism of DISH and the specific characteristics underlying cell signaling.

Answer: We thank the reviewer for the question/comment. We and others have found evidence or hypothesized that spinal fusion may be hindered by the secretion of BMP inhibitors, which might be secreted from IVD cells, i.e. see for instance Kok D, et al. (2019) PLoS ONE 14(4):e0215536. doi: 10.1371/journal.pone.0215536) and Makino et al. (2018) Int J Mol Sci 19(8): doi: 10.3390/ijms19082430 and  Brown et al. (2018) Cartilage 1947603518754628. doi: 10.1177/1947603518754628. Thus, considering that DISH is a metabolic disease where bigger areas of the neighborhood of the spine is ossified for yet unidentified causes but the intervertebral discs seem to be resistant to this cell signalling seem to further support our working hypothesis that IVD cells secrete a considerable amount and array of various BMP antagonists. For this we and others have recently found experimental evidence. Thus, we added more arguments to the introduction why we consider this research to be important to understand metabolic and osteogenic diseases.

2.4 Why are the Materials and Methods section (section 4) reported after Results and Discussion? Please move them to section 2 and move the results and discussion down correspondingly.

Answer: We have now re-consulted the author guidelines of Applied Sciences. Indeed the order of the subsections can be kept in the traditional way, which differs from other journals of MDPI, such as IMJS, where it is the case that M&M is the last section. Thus, we rearranged the order and also think that the results will be much better understood in this classical order.

2.5 The discussion section is also quite lengthy, and repetitious in parts. For example, lines 8-9 are somewhat repeated in lines 12-13 (page 7)

Answer: We have shortened the discussion according to the recommendation.

Reviewer 3 Report

In the following review I will address major observations, and I will require to see this manuscript again for further evaluation.

Author Response

Reviewer 3

3.1 The article represents an advance in the knowledge of a relevant clinical problem. Through gene expression analysis by using qPCR and immunofluorescence the article shows changes in gene expression of key bone signalling proteins on the cell populations of the intervertebral disk of patients with DISH. The article has the significant value and challenge of working with primary cells extracted from intervertebral disc of patietns with DISH and a control group. This manuscript is part of an active research project that has made significant contributions in the field. However, the presentation of the data requires major improvements, as well as the justification of the study, in order to provide the readers, the maximum utility of the findings here described. Similarly, there are issues in technical writing and methods descriptions that need to be addressed before pursuing a thorough editing.

Answer: We thank the reviewer for the appreciaiton of our work.

3.2 Abstract: Please add that different areas of the IVD are analyzed in the study.

Answer: We have added the details to the abstract.

Introduction:

3.3 It provides sufficient information for understanding the clinical problem; However, it lacks information of TGF-b/ BMP biology and function, and how they are related to physiological and pathological calcification process. Importantly, there is no justification to focus a study on BMP/TGF-b, other than one sentence and one reference [24], which is not even highlighted as relevant factors but referred just as “other growth factors…” (line 41) If there is no rationale, why should the reader continue reading this article?

Answer: We have addressed this gap in the introduction in the current revision. We have added additional information on the TGFb/BMP pathway.

Results:

3.4 For journals like Applied Sciences, that are formatted with materials and methods at the end of the research manuscript, it is necessary to introduce the sub sections of the results with a brief description of the experiment performed. It is not just cut and paste the methods at the end, then it should be read 1 intro-4 methods- 2results – 3 Discussion as n regular articles. For example, in section 2.1. line 15. Is it tissue gene expression? Cells? a couple of sentences would help with the flow of the article here.

Answer: Please, see 2.4

3.5 By convention when we talk about genes it should be italicised, when talking about protein names go in regular font.

Answer: We thank the reviewer to point out the nomenclature rules for genes versus proteins. We went through the manuscript and adjusted this accordingly.

3.6 Figure 1: X and Y axis, fonts are not readable, must be changed. Is it possible to pool data from the relevant genes and present them in a different chart? Dot plot or column plot? I can see a few genes that are highly express in across the samples, in that way the relevance and differences of specific genes would be highlighted, and easier to understand by the reader. It is clear to this observer that some genes are repeatedly up regulated and others down regulated. If it doesn’t help in the main document at least submitting them as supplement would help

Answer: We have enlarged the font size of all the axes in the scatterplot figure.  As suggested we have moved this figure now to the supplementary file material labelled as “supplementary figure 1”. We have produced a new figure 1 illustrating the normalized gene expression at eight selected genes, where disregulations were found in the two defined groups.

Figure 2:

3.7 Color coding the sample in a heat-map is a good idea, help to visualize the differences. The second line of colour codding could be added to identify NPC, AFC or CEPC. CP1CEP sample is miss labelled in the second part of the phenogram. Organizing genes in alphabetic order do not allow to observe similarities/differences in the chart. I would recommend performing a bidimensional heatmap that helps organizing genes according to their similarities in distribution as well, then similarities/differences would become more evident in the chart.

Answer: We thank the reviewer for this idea. We have added the additional colour-coding as suggested to distinguish the three different tissue types. We also have corrected the wrongly colour-coded sample. However, the software package that was used to derive the heatmap, i.e., the CFX96 3.1 manager software does not allow the option to sort the genes according to their similarities. Thus, we decided to use the order from the Bio-Rad software analysis package.

3.8 The methods indicate that quantification of gene expression in the chart is presented in colour code as up-regulated or down-regulated with a threshold of 4-fold. However, there is a gradient of green and red along the chart and not a discrete distribution of red-black-green. This need clarification in the methods section.

Answer: The procedure follows the manual and software implented from Bio-Rad, https://www.bio-rad.com/webroot/web/pdf/lsr/literature/10010424.pdf on p. 121: A legend below the heat map shows the range of normalized expression, which corresponds to the following:• Upregulation (red). Higher expression• Downregulation (green). Lower expression• No change (black). “The lighter the shade of color, the greater the relative normalized expression difference. If no normalized expression value can be calculated, the square will be black with a white X.”

Section 2.2

3.9 Page 5, Line 10. Starting a section with Furthermore is not appropriated

 Answer: This was rephrased.

Figure 3:

3.9. It is difficult to obtain good immunofluorescence images from secretion molecules such as BMP or IGF. However, most images would benefit from further processing to increase contrast. It is easy nowadays with a huge variety of image processing software available. Please remove the text from inside the pictures, it is redundant and does not read well.The legend refers to the technique as fluorescent immune histology. Histology refers to the study of tissues. It could be a great contribution to have the histological staining, but the figure is only immunofluorescence or immunocytochemistry as the heading of the section indicates. If available, I encourage to ad figure with histological sections of the samples. This reviewer understands that it may not be available since they were used for cell extraction, thus this is just a suggestion.

Answer: we unfortunately are not in possession of clinical material of good enough quality for histological sections. We have improved the labelling/quality overall for these microscopy pictures.

Materials and methods

Section 4.2

Well explained the wet lab process of gene expression analysis. But lack of details on the statistical analysis. Does not mention the software used on the analysis, or functions. How was the phenogram made? Is it an unsupervised hierarchical clustering?

Answer: The presented cluster gram shows the data in a hierarchy based on the degree of similarity of expression for different targets and samples. So, by this, the reviewer is correct it is unsupervised hierarchical clustering. The software to generate the clustering and heat-map was based on the integrated analysis tool of the CFX96 Touch™ PCR machine, i.e., Bio-Rad CFX manager 3.1 from Bio-Rad. We have added this information to the material  and methods

Round 2

Reviewer 1 Report

Authors responded to my comments.

Author Response

We thank the reviewer for this comment.

Author Response

Answer: We thank the reviewer for the specifications and her/his time to improve our manuscript prior to publication. Please, find below our point-to-point-rebuttal letter.

  • In general, still needs a more defined As I see from here the intended structure is:Paragraph 1: definition and epidemiology 2. clinical aspects – anatomical description 3-4 histological description / metabolomics-genes expression 5 define aims of this study.I can read the intention of this type of structure but is not clear. There are mixed sentences on each of them and thus the content of each paragraph is not completely clear.

Answer: We have now rewritten almost the entire introduction section according to the suggestion of reviewer 1 to improve the text flow and the readability as suggested.

  • Page 1 Lines 40-44: Although refers to increasing rates with gender and age, it does not refer to the number of people affected, or prevalence in some age groups. Any of those numbers will help understanding the magnitude of the problem. Page 2 Lines 1-7: A description of clinical significance is missing: e.g. does it generate pain?, discomfort? As it is described seems that most patients are asymptomatic, but a few has spine instability, then is not a real problem. If there is no clinical consequence, why to study this?

Answer: We have added now the etiologies of the different age groups to understand the magnitude of the problem. We also have added that the condition may be affiliated with strong and consistent back pain and is often co-diagnosed with osteoarthritis. DISH patients need special treatment and cause high costs to the society.

  • Lines 7-10: Why a sentence about epidemiology here? I thought it was the theme of the previous paragraph.

Answer: We have moved this sentence. Please, see our answer above.

  • Line 13: Paragraph starts again with an anatomical description and the following paragraph is about molecular pathways?

Answer: We have rearranged the logics and the flow of the introduction.. Please, see our answer above.

  • Line 14 delete the comma before “that” Line 16 : mentioned “starts with molecular pathways, continues with cadaveric studies to finish the paragraph with BMPs” Line 26 is it a new Paragraph?

Answer: We have completely rearranged the logics and the flow of the introduction. Please, see our answer above.

Methods:

 Well described and with proper detail to reproduce the experiments. Page 7, Line 3 : Never heard of “low passage” I’m not sure if it’s a proper term. Keep it just as P1/P2

Answer: we changed this as suggested.

Results:

 Page 7 line 22: provide the information higher to lower, it seems that IL-6 shows higher differences in fold change and p-value so should go first. The “p” of p-value should not be capitalized

Answer: we changed the formatting of the “P”- of the P-values into plain font.

 Page 8 line 1: Was IGF1 up-regulated? p-value of 0.17 means no difference. Page 8 line 3: provide fold change and p-value for GDF5 and 6

 Answer: IGF-1 tended to be up-regulated. However, due to the high variance this effect was not significant. Thus, we weakened the statements in the results section and expressed this as a trend throughout the manuscript. However, we kept it in the title of the manuscript as this was in some DISH donors really dramatically upregulated.

Page 7 line 26: “Furthermore, IGF1 tended to be up-regulated in DISH-IVD donors (174.1 ± 120.6-fold, P = 0.1704). Growth and Differentiation Factor 5 (GDF5) and GDF6 both tended to be down-regulated in the DISH-IVDs (i.e., -11.5 ± 10.0, P = 0.26 and -3.7 ± 3.1 P = 0.30, respectively, Figure 1, supplementary Figure 1).”

 Figure 3: As in the previous round I asked for higher contrast on the images since the staining was not observed in the figure, now several images are saturated, which is not acceptable. To date, there are several image processing software in the market that would help with the proper presentation of these images, I highly suggest revise histograms and determine thresholds per channel to provide informative images about the immunolocalization of the studied proteins.

Answer: We thank the reviewer for this comment. We re-edited the settings of these images and have used the exact same imaging parameters to make sure we modified all images alike. We also ensured that no saturation of pixels happend in some of the images.

 Discussion:

Similar problems than intro, about clarity of the paragraphs and sections. Also issues with the verve tense, uses past continuous in several sections referring to article on the past or either this same article: “we were detecting” “could be shown” In this section I would recommend a technical English advice.

 Page 11 Paragraph 1 starts with description of findings, continue with limitation on sample collection, in line 7-8 refers to IGF-1 and in line 10 goes back to sample collection.

Answer: we went through the discussion and carefully checked for proper English grammar and simple past usage. We also intensively worked on the flow and the English grammar.

 Line 1: you can report or you do report?

 Answer: This has been changed.

Line 6: “a lot” is not an academic term.

Answer: This has been removed or changed.

Line 19-20: That’s introduction Line 21: Should be the beginning of the discussion. In a constant error in this article, it goes back to first line of discussion, which makes no clear what paragraph is about what.

Lines 27-30: Inconsistence on citation style: A reference that should have been located after the first sentence should be “calcified trabeculli [32]” - (line 28) is pulled back to the end of the net sentence on the justification about the same article. However, in the following sentence the citation is located immediately after mentioning the article.

Answer: We have unified the citation style throughout the manuscript.

32: is UMR106 an osteoblastic cell line? Please clarify in the text.

Answer: Page 11 line 32: We have modified the text to “the osteosarcoma-derived UMR106 cell line».

33: avoid the word “never”.

Answer: We checked the text for the wording and replaced it.

35-42 It could be shown is not the beginning of a sentence: Who show/demonstrates what? “Could” is not a descriptive term. It has been shown, it is known, it has been demonstrated. …

Answer: This was corrected.

  1. Yakar et al… if the study is from 2002 is past, not past continuous.

Answer: This was corrected.

Round 3

Reviewer 3 Report

General comments:

The authors have made substantial improvements to the article, that made it suitable for publication. Congratulations and good luck in future research in this project.

Below I specify minor changes to be made.

Page 2, lines 1-2: change one “furthermore” for another connector.

              14-19: change one of the “Until today” for another expression

Page 7: line 8 typo in “permeabilized”

 Page 11:                    21: “trend” does not sound specific, please be more precise here with what was the observation.

                     28 : Phenomena?

Page 12: Furthermore and also are redundant

               Conclusions need to be made on statistical facts. Delete those that are not conclusive because they could not be conclusions.

                  Conclusions should not start with transitional words. They are independent statements.

Author Response

Rebuttal Letter

We thank the reviewer again for taking the time to read our work carefully and to improve our work. This is highly appreciated. We carefully re-went through the entire text and made additional improvements to the text flow and grammar.

Below are our answers for the few minor points which needed to be addressed.

The authors have made substantial improvements to the article that made it suitable for publication. Congratulations and good luck in future research in this project.

Answer: We thank the reviewer for this comment.

Page 2, lines 1-2: change one “furthermore” for another connector.

Answer: We changed this with “moreover”.

14-19: change one of the “Until today” for another expression

Answer: We changed this.

Page 7: line 8 typo in “permeabilized”

Answer: We changed this.

Page 11: line 21: “trend” does not sound specific. Please be more precise here with what was the observation.

Answer: We have better focused on the observation.

28 : Phenomena?

Answer: We have better focused on the observation.

Page 12: Furthermore and also are redundant

Answer: We changed this.

Conclusions need to be based on statistical facts. Delete those that are not conclusive because they could not be conclusions.

Conclusions should not start with transitional words. They are independent statements.

Answer: We deleted the conclusions that were weak and removed the transition words as suggested.